# DYNAMIC BACKDOOR ATTACKS AGAINST DEEP NEURAL NETWORKS

## ABSTRACT

Current Deep Neural Network (DNN) backdooring attacks rely on adding static triggers (with fixed patterns and locations) on model inputs that are prone to detection. In this paper, we propose the first class of dynamic backdooring techniques: Random Backdoor, Backdoor Generating Network (BaN), and conditional Backdoor Generating Network (c-BaN). Triggers generated by our techniques have random patterns and locations. In particular, BaN and c-BaN based on a novel generative network are the first two schemes that algorithmically generate triggers. Moreover, c-BaN is the first conditional backdooring technique that given a target label, it can generate a target-specific trigger. Both BaN and c-BaN are essentially a general framework which renders the adversary the flexibility for further customizing backdoor attacks. We extensively evaluate our techniques on three benchmark datasets and show that our techniques achieve almost perfect attack performance on backdoored data with a negligible utility loss. More importantly, our techniques can bypass state-of-the-art defense mechanisms.

## 1 INTRODUCTION

Recent research has shown that deep neural network (DNN) models are vulnerable to various security and privacy attacks (Papernot et al., 2016; 2017; Shokri et al., 2017; Salem et al., 2019; 2020; Tramèr et al., 2016; Oh et al., 2018). One such attack that receives a large amount of attention is *backdoor*, where an adversary trains a DNN model which can intentionally misclassify any input with an added *trigger* (a secret pattern constructed from a set of neighboring pixels) to a specific *target label*. Backdoor attacks can cause severe security consequences. For instance, an adversary can implant a backdoor in an authentication system to grant herself unauthorized access.

Existing backdoor attacks generate static triggers, in terms of fixed trigger pattern and location (on the model input). For instance, Figure 1a shows an example of triggers constructed by Bad-Nets (Gu et al., 2017), one popular backdoor attack method, on the CelebA dataset (Liu et al., 2015).

As we can see, BadNets in this case uses a white square as a trigger and always places it in the top-left corner of an input. This static nature of triggers has been leveraged to create most of the current defenses against the backdoor attack (Wang et al., 2019; Liu et al., 2019a; Gao et al., 2019).

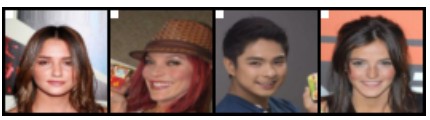

(a) Static backdoor

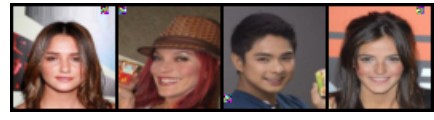

(b) Dynamic backdoor

Figure 1: A comparison between static and dynamic backdoors on CelebA.

In this paper, we propose the first class of backdooring techniques against deep neural network models that generate dynamic triggers, in terms of trigger *pattern* and *location*. We refer to our techniques as *dynamic backdoor attacks*. Figure 1b shows an example. Dynamic backdoor attacks offer the adversary more flexibility, as they allow triggers to have different patterns and locations. Moreover, our techniques largely reduce the efficacy of the current defense mechanisms demonstrated by our empirical evaluation. In addition, we extend our techniques to work for all labels of the backdoored DNN model, while the current backdoor attacks only focus on a single or a few target labels. This further increases the difficulty of our backdoors being mitigated.

In total, we propose 3 different dynamic backdoor techniques, i.e., *Random Backdoor*, *Backdoor Generating Network (BaN)*, and *conditional Backdoor Generating Network (c-BaN)*. In particular, the latter two attacks algorithmically generate triggers to mount backdoor attacks which are first of their kind.

To demonstrate the effectiveness of our proposed techniques, we perform empirical analysis with three DNN model architectures over three benchmark datasets. All of our techniques achieve almost a perfect backdoor accuracy, i.e., the accuracy of the backdoored model on the backdoored data is approximately 100%, with a negligible utility loss. Moreover, we show that our techniques can bypass three state-of-the-art backdoor defense techniques, namely Neural Cleanse (Wang et al., 2019), ABS (Liu et al., 2019a), and STRIP (Gao et al., 2019). In general, our contributions can be summarized as follows: 1) We broaden the class of backdoor attacks by introducing the dynamic backdoor attacks. 2) We propose both BaN and c-BaN, which are the first algorithmic backdoor paradigm. 3) Our dynamic backdoor attacks achieve strong performance while bypassing the current state-of-the-art backdoor defense techniques.

## 2 RELATED WORK

**Backdoor Attacks:** Gu et al. (Gu et al., 2017) introduce BadNets, the first backdoor attack on DNN models. BadNets uses the MNIST dataset and a square-like trigger with a fixed location, to show the applicability of the backdoor attacks in the DNN settings. Liu et al. (Liu et al., 2019b) later propose a more advanced backdooring technique, namely the Trojan attack. They simplify the threat model of BadNets by eliminating the need for access to the training data used to train the target model. The main difference between these two attacks (BadNets and Trojan attacks) and our work is that both attacks only consider static backdoors in terms of triggers' pattern and location. Our work extends the backdoor attacks to consider dynamic patterns and locations of the triggers. We focus on backdoor attacks against image classification models, but backdoor attacks can be extended to other scenarios, such as Federated Learning (Wang et al., 2020), Video Recognition (Zhao et al., 2020), Transfer Learning (Yao et al., 2019), and Natural language processing (NLP) (Chen et al., 2020).

To increase the stealthiness of the backdoor, Saha et al. (Saha et al., 2020) propose to transform the backdoored images into benign-looking ones, which makes them harder to detect. Lie et al. (Liu et al., 2020) introduce another approach, namely, the reflection backdoor (Refool), which hides the triggers using mathematical modeling of the physical reflection property. Another line of research focuses on exploring different methods of implementing backdoors into target models. Rakin et al. (Rakin et al., 2020) introduce the Targeted Bit Trojan (TBT) technique, which instead of training the target model, flips some bits in the target models' weights to make it misclassify all the inputs. Tang et al. (Tang et al., 2020) present a different approach, where the adversary appends a small Trojan module (TrojanNet) to the target model instead of fully retraining it.

**Defenses Against Backdoor Attacks:** Wang et al. (Wang et al., 2019) propose Neural Cleanse (NC), a backdoor defense method based on reverse engineering. For each output label, NC tries to generate the smallest trigger, which converts the output of all inputs applied with this trigger to that label. NC then uses anomaly detection to find if any of the generated triggers are actually a backdoor or not. Later, Liu et al. (Liu et al., 2019a) propose another model-based defense, namely, ABS. ABS detects if a target model contains a backdoor or not, by analyzing the behaviour of the target model's inner neurons when introducing different levels of stimulation. Also, Gao et al. (Gao et al., 2019) propose STRIP, a backdoor defense method based on manipulating the input, to find out if it is backdoored or not. More concretely, STRIP fuses the input with multiple clean data, one at a time. Then it queries the target model with the generated inputs, and calculate the entropy of the output labels. Backdoored inputs tend to have lower entropy than the clean ones.

Besides the above, there are multiple other types of attacks against deep neural network models, such as adversarial examples (Vorobeychik & Li, 2014; Carlini & Wagner, 2017; Li & Vorobeychik, 2015; Tramèr et al., 2017; Xu et al., 2018), poisoning attack (Jagielski et al., 2018; Suciu et al., 2018; Biggio et al., 2012), and property inference (Ganju et al., 2018; Melis et al., 2019).

## 3 DYNAMIC BACKDOORS

### 3.1 THREAT MODEL

The dynamic backdoor attack is a training time attack, i.e., the adversary is the one who trains the backdoored DNN model. To implement our attack, we assume the adversary controls the training of the target model and has access to the training data following previous works (Gu et al., 2017; Yao et al., 2019; Chen et al., 2020). To launch the attack (after publishing the model), the adversary first adds a trigger to the input and then uses it to query the backdoored model. This can happen either digitally, where the adversary digitally adds the trigger to the image, or physically, where the adversary prints the trigger and places it on the image. This added trigger makes the backdoored model misclassify the input to the target label.

### 3.2 RANDOM BACKDOOR

We start by presenting our Random Backdoor technique. Abstractly, the Random Backdoor technique constructs triggers by sampling them from a uniform distribution, and adding them to the inputs at random locations. We first introduce using Random Backdoor to implement a dynamic backdoor for a single target label, then we generalize it to multiple target labels.

**Single Target Label:** In this setting, we construct a set of triggers ($\mathcal{T}$) and a set of possible locations ($\mathcal{K}$), such that for any trigger sampled from $\mathcal{T}$ and added to any input at a random location sampled from $\mathcal{K}$, the model will output the specified target label. More formally, for any location $\kappa_i \in \mathcal{K}$, any trigger $t_i \in \mathcal{T}$, and any input $x_i \in \mathcal{X}$: $\mathcal{M}_{bd}(\mathcal{A}(x_i, t_i, \kappa_i)) = \ell$, where $\ell$ is the target label, $\mathcal{T}$ is the set of triggers, $\mathcal{K}$ is the set of locations, $\mathcal{M}_{bd}$ is the backdoored model, and $\mathcal{A}$ is the backdoor adding function (it adds the trigger $t_i$ to the input $x_i$ at the location $\kappa_i$).

To implement such a backdoor, an adversary needs to first select her desired trigger locations and create the set of possible locations $\mathcal{K}$. Then, she uses both clean and backdoored data to update the model for each epoch similar to the BadNets technique but with two differences. First, instead of using a fixed trigger for all inputs, each time the adversary wants to add a trigger to an input, she samples a new trigger from a uniform distribution, i.e., $t \sim \mathcal{U}(0, 1)$. Here, the set of possible triggers $\mathcal{T}$ contains the full range of all possible values for the triggers, since the trigger is randomly sampled from a uniform distribution. Second, instead of placing the trigger in a fixed location, she places it at a random location $\kappa$, sampled from the predefined set of locations, i.e., $\kappa \in \mathcal{K}$. Note that Random Backdoor is not only limited to uniform distribution, other distributions, such as Gaussian distribution, can be used as well.

**Multiple Target Labels:** Next, we consider the more complex case of having multiple target labels. Without loss of generality, we consider implementing a backdoor for each label in the dataset. This means that for any label $\ell_i \in \mathcal{L}$, there exists a trigger $t$ which when added to the input $x$ at a location $\kappa$, will make the model $\mathcal{M}_{bd}$ output $\ell_i$. More formally, $\forall \ell_i \in \mathcal{L} \, \exists \, t, \kappa : \mathcal{M}_{bd}(\mathcal{A}(x, t, \kappa)) = \ell_i$. To achieve the dynamic backdoor behaviour, each target label should have a set of possible triggers and a set of possible locations.

We generalize the Random Backdoor technique by dividing the set of possible locations $\mathcal{K}$ into disjoint subsets for each target label, while keeping the trigger construction method the same as in the single target label case, i.e., the triggers are still sampled from a uniform distribution. For instance, for the target label $\ell_i$, we sample a set of possible locations $\mathcal{K}_i$, where $\mathcal{K}_i$ is subset of $\mathcal{K}$ ($\mathcal{K}_i \subset \mathcal{K}$). The adversary can construct the disjoint sets in two steps. First, the adversary selects all possible triggers locations and constructs the set $\mathcal{K}$. Second, for each target label $\ell_i$, she constructs the set of possible locations for this label $\mathcal{K}_i$ by sampling the set $\mathcal{K}$. Then, she removes the sampled locations from the set $\mathcal{K}$. We propose a simple algorithm to assign the locations for the different target labels. Concretely, we uniformly split the image into non-intersecting regions, and assign a region for each target label, in which the triggers' locations can move vertically. Figure 2 shows an example of our location setting technique for a use case with 6 target labels. We stress that this is one way of dividing the location set $\mathcal{K}$ to the different

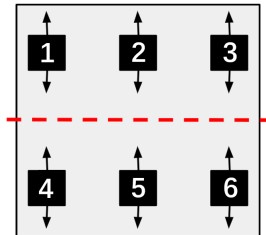

Figure 2: An illustration of our location setting technique for 6 target labels.

target labels. However, an adversary can choose a different way of splitting the locations inside $\mathcal{K}$ to the different target labels. The only requirement the adversary has to fulfill is to avoid assigning a location for different target labels. Later, we will show how to overcome this limitation with our more advanced c-BaN technique.

### 3.3 BACKDOOR GENERATING NETWORK (BaN)

Our Random Backdoor technique successfully implements dynamic triggers, however, it offers the adversary limited flexibility as triggers are sampled from a fixed distribution. Moreover, the triggers are sampled independently of the target model. In other words, the Random Backdoor technique does not search for the best triggers to implement the backdoor attack. To address these limitations, we introduce our second technique to implement dynamic backdoors, namely, BaN, the first approach to algorithmically generate backdoor triggers.

BaN is inspired by the state-of-the-art generative model, i.e., GANs (Goodfellow et al., 2014). It trains a generator with the target model as the discriminator to learn the best patterns for a backdoor trigger. After training, BaN can generate a trigger ($t$) for each noise vector ($z \sim \mathcal{U}(0,1)$). This trigger is then added to an input using the backdoor adding function $\mathcal{A}$, to create the backdoored input as shown in Figure 3a. Similar to the previous approach (Random Backdoor), the generated triggers are placed at random locations.

**Single Target Label:** We start with presenting how to implement a dynamic backdoor for a single target label using BaN. First, the adversary creates the set $\mathcal{K}$ of the possible locations. She then jointly trains a BaN with the backdoored $\mathcal{M}_{bd}$ model as follows:

1. The adversary starts each training epoch by querying the clean data to the backdoored model $\mathcal{M}_{bd}$. Then, she calculates the clean loss $\varphi_c$ between the ground truth and the output labels. We use the cross-entropy loss for our clean loss.

2. She then generates $n$ noise vectors, where $n$ is the batch size.

3. On the input of the $n$ noise vectors, the BaN generates $n$ triggers.

4. The adversary then creates the backdoored data by adding the generated triggers to the clean data using the backdoor adding function $\mathcal{A}$.

5. She then queries the backdoored data to the backdoored model $\mathcal{M}_{bd}$ and calculates the backdoor loss $\varphi_{bd}$ on the model's output and the target label. Similar to the clean loss, we use the cross-entropy loss as our loss function for $\varphi_{bd}$.

6. Finally, the adversary updates the backdoor model $\mathcal{M}_{bd}$ using both the clean and backdoor losses ($\varphi_c + \varphi_{bd}$) and updates the BaN with the backdoor loss ($\varphi_{bd}$).

**Multiple Target Labels:** We now consider a more complex case of building a dynamic backdoor for multiple target labels using BaN. To recap, BaN generates general triggers and not label specific triggers. In other words, the same trigger pattern can be used to trigger multiple target labels. Thus similar to Random Backdoor, we depend on the location of the triggers to determine the output label.

We follow the same approach of the Random Backdoor technique to assign different locations for different target labels (Section 3.2), to generalize the BaN technique. More concretely, the adversary starts by creating disjoint sets of locations for all target labels. Next, she follows the same steps as in training the backdoor for a single target label, while repeating from step 2 to 5 (see above) for each target label and adding all their backdoor losses together. More formally, for the multiple target label case the backdoor loss is defined as: $\sum_i^{|\mathcal{L}'|} \varphi_{bd_i}$, where $\mathcal{L}'$ is the set of target labels, and $\varphi_{bd_i}$ is the backdoor loss for target label $\ell_i$.

### 3.4 CONDITIONAL BACKDOOR GENERATING NETWORK (c-BaN)

Both Random Backdoor and BaN have the limitation of not having label specific triggers and only depending on the trigger location to determine the target label. We now introduce our third and most advanced technique, the c-BaN, which overcomes this limitation. With c-BaN, any location $\kappa$ inside $\mathcal{K}$ can be used to trigger any target label. To achieve this location independency, the triggers need to

Figure 3: An overview of the BaN and c-BaN techniques.

be label specific. Therefore, we add the target label as an additional input to BaN for conditioning it to generate target specific triggers.

We construct c-BaN by adding an additional input layer to BaN to include the target label as an input. Figure 3b represents an illustration of c-BaN. As the figure shows, the noise vector and the target label are encoded to latent vectors with the same size (to give equal weights for both inputs). These two latent vectors are then concatenated and used as an input to the next layer. Here, we use one-hot encoding to encode the target label. c-BaN is trained similarly to BaN with one exception, that is the adversary does not have to create disjoint sets of locations for all target labels, she can use the complete location set $\mathcal{K}$ for all target labels.

To use c-BaN, the adversary first samples a noise vector and one-hot encodes the label. Then, she inputs both of them to c-BaN to generate a trigger. The adversary uses the backdoor adding function $\mathcal{A}$ to add the trigger to the target input. Finally, she queries the backdoored input to the backdoored model, which will output the target label.

One major advantage of BaN and c-BaN is their flexibility, i.e., they allow the adversary to customize her backdoor by adapting the loss functions. For instance, the adversary can adapt to different defenses against the backdoor attack that can be modeled as deep neural network models. This can be achieved by adding any defense as a discriminator into the training of BaN or c-BaN. Adding this discriminator will penalize/guide the backdoored model to bypass the modeled defense.

## 4 EVALUATION

### 4.1 EXPERIMENTAL SETUP

We utilize three image datasets to evaluate our techniques, including MNIST, CelebA, and CIFAR-10. For the CelebA dataset, we first scale the images to $64 \times 64$, and select the top three most balanced attributes, i.e., Heavy Makeup, Mouth Slightly Open, and Smiling. Then, we concatenate them into $8$ classes to create a multiple label classification task. We use these three datasets since they are widely used as benchmark datasets for various security/privacy and computer vision tasks.

The architectures for our target models, BaN, and c-BaN are presented in Section A.1. For evaluating the dynamic backdoor attack's performance, we define the following two metrics: *Backdoor success rate* which calculates the backdoored model's accuracy on the backdoored data; *Model utility* which measures the original functionality of the backdoored model. We quantify the model utility by comparing the accuracy of the backdoored model with the accuracy of a clean model on clean data. Closer accuracies imply a better model utility. Note that for all of our techniques, the backdoor success rate is almost always 100%, thus we mainly focus on model utility. All of our experiments are implemented using Pytorch and our code will be published for reproducibility.

### 4.2 RANDOM BACKDOOR

**Single Target Label:** We split each dataset into training and testing datasets. The training datasets are used to train the MNIST and CelebA models from scratch, and fine-tune a pre-trained VGG-19 model for CIFAR-10. We use the testing dataset as our clean testing dataset and construct a backdoored testing dataset, by adding triggers to all samples of the testing dataset. The former is used to measure model utility while the latter is used to measure the attack success rate.

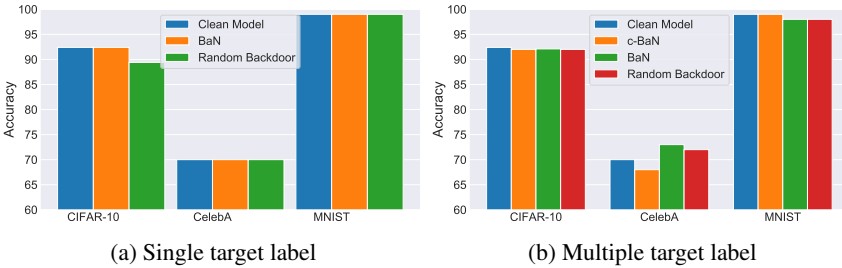

(a) Single target label  (b) Multiple target label

Figure 4: The result of our dynamic backdoor techniques for single and multiple target label on the clean testing dataset.

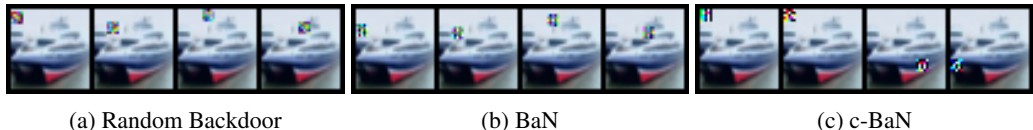

(a) Random Backdoor  (b) BaN  (c) c-BaN

Figure 6: Visualization result of our techniques for the first four labels of the CIFAR-10 dataset.

As (Figure 4a) shows, Our backdoored models achieve the same performance as clean models for the MNIST and CelebA datasets. For CIFAR-10, there is a slight drop in performance (about 2%). This shows that our Random Backdoor can implement a perfectly functioning backdoor, i.e., the backdoor success rate of $\mathcal{M}_{bd}$ is 100%, with a negligible utility loss. Figure 5a shows some qualitative examples of backdoored data generated by Random Backdoor. As expected, the triggers look distinctly different and are located at different locations.

**Multiple Target Labels:** We use a similar evaluation setting to the single target label case with one exception. To evaluate the attack success rate with multiple target labels, we construct a backdoored testing dataset for each target label by generating and adding triggers to the clean testing dataset. In other words, we use all images in the testing dataset to evaluate all possible labels. Even in this case, we still achieve nearly perfect attack success rate.

Figure 4b show the model utility in this case. Using our Random Backdoor technique, we are able to train backdoored models that achieve similar performance as the clean models for all datasets. For instance, for the CIFAR-10 dataset, our Random Backdoor technique achieves 92% accuracy, which is very similar to the accuracy of the clean model (92.4%). For the CelebA dataset, the Random Backdoor technique achieves a slightly (about 2%) better performance than the clean model. We believe this is due to the regularization effect of the Random Backdoor technique. Finally, for the MNIST dataset, both models achieve similar performance with just 1% difference between the clean model (99%) and the backdoored one (98%). Figure 6a further shows some examples of backdoored images by Random Backdoor in the multi target label setting. Moreover, we further visualize in Figure 7a the dynamic behavior of the triggers generated by our Random Backdoor technique. Without loss of generality, we generate triggers for the target label 5 (plane) and add them to randomly sampled CIFAR-10 images. As Figure 7a shows, the generated triggers have different patterns and locations for the same target label, which achieves our desired dynamic behavior.

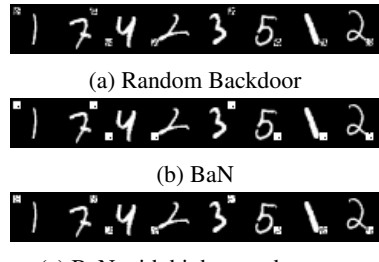

(a) Random Backdoor

(b) BaN

(c) BaN with higher randomness

Figure 5: Visualisation of the dynamic behaviour for our techniques for a single target.

### 4.3 BACKDOOR GENERATING NETWORK (BAN)

Next, we evaluate our BaN technique. We follow the same settings for the Random Backdoor technique, except with respect to how the triggers are generated. We train our BaN model and generate the triggers as mentioned in Section 3.3.

**Single Target Label:** Similar to the Random Backdoor, the BaN technique achieves perfect backdoor success rate with a negligible utility loss. As shown in Figure 4a, our BaN backdoored models achieve 99%, 92.4% and 70% accuracy on the MNIST, CIFAR-10, and CelebA, respectively, which is the same performance of the clean models.

We visualize the BaN generated triggers using the MNIST dataset in Figure 5b. The generated triggers look very similar as shown in Figure 5b. This behaviour is expected as the MNIST dataset is simple, and the BaN technique does not have any explicit loss to enforce the network to generate different triggers. However, to show the flexibility of our approach, we increase the randomness of the BaN network by simply adding one more dropout layer after the last layer, to avoid the overfitting of the BaN model to a unique pattern. We show the results of the BaN model with higher randomness in Figure 5c. The resulting model still achieves the same performance, i.e., 99% model utility and 100% backdoor success rate, but as the figure shows the triggers look significantly different. This again shows that our framework can easily adapt to the requirements of an adversary.

**Multiple Target Labels:** Our BaN backdoored models are able to achieve almost the same accuracy as the clean model for all datasets, as shown in Figure 4b. For instance, for the CIFAR-10 dataset, our BaN achieves 92.1% accuracy, which is only 0.3% less than the performance of the clean model (92.4%). Similar to the Random Backdoor backdoored models, our BaN backdoored models achieve a marginally better performance for the CelebA dataset. More concretely, our BaN backdoored models trained for the CelebA dataset achieve about 2% better performance than the clean model. We also believe this improvement is due to the regularization effect of the BaN technique. Finally, for the MNIST dataset, our BaN backdoored models achieve strong performance on the clean testing dataset (98%), which is just 1% lower than the performance of the clean models (99%).

Similar to the Random Backdoor, we visualize the results of the BaN backdoored models with two figures. The first (Figure 6b) shows the different triggers for the different target labels on the same CIFAR-10 image, and the second (Figure 7b) shows the different triggers for the same target label (plane) on randomly sampled CIFAR-10 images. As both figures show, the BaN generated triggers achieve dynamic behaviour in both the location and patterns.

### 4.4 CONDITIONAL BACKDOOR GENERATING NETWORK (C-BAN)

Next, we evaluate our c-BaN technique. For the single target label case, the c-BaN technique is the same as the BaN technique. Thus, we only consider the multiple target labels case here. We follow a similar setup as introduced in Section 4.3, with the exception of following Section 3.4 to train and evaluate $\mathcal{M}_{bd}$. For the set of possible locations $\mathcal{K}$, we set it to 4.

Our evaluation shows that c-BaN is able to build models with good model utility (see Figure 4b). For instance, for the CIFAR-10 and MNIST datasets, our c-BaN achieves 92%, and 99% accuracy which is very similar to the accuracy of the clean models, i.e., 92.4% and 99%, respectively.

We compare the dynamic behaviour of our three different techniques using two different figures. First, Figure 6 shows the visualization of backdooring a single image to the first four labels of the CIFAR-10 dataset. As the figure shows, the Random Backdoor Figure 6a has the most random patterns, which is expected as they are sampled from a uniform distribution. The figure also shows the different triggers' patterns and locations used for the different techniques.

Second, Figure 7 visualizes the different triggers' patterns and locations for the same target label. As the figure shows, the Random Backdoor (Figure 7a) and BaN (Figure 7b) generated triggers can move vertically, however, they have a fixed position horizontally as mentioned in Section 3.2. The c-BaN (Figure 7c) triggers also show different locations. However, the locations of these triggers are more distant and can be shared for different target labels, unlike the other two techniques. Furthermore, the figure shows that most of our triggers have different patterns for our techniques for the same

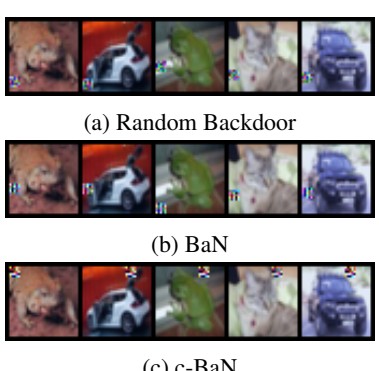

(a) Random Backdoor

(b) BaN

(c) c-BaN

Figure 7: Visualisation of triggers generated for the same target.

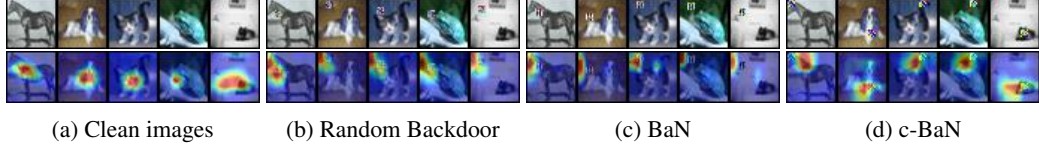

| (a) Clean images | (b) Random Backdoor | (c) BaN | (d) c-BaN |

Figure 8: Visualization of attention maps for all our techniques using the Grad-CAM technique.

target label, which achieves our targeted dynamic behavior concerning the patterns and locations of the triggers.

Finally, we compare the attention of the backdoored models on both clean and backdoored inputs. We use the Gradient-weighted Class Activation Mapping (Grad-CAM) technique (Selvaraju et al., 2017) to compute the attention maps for our backdoored models. These maps show the most influential parts of the input that resulted in the model's output. Figure 8 depicts the results of our three different techniques. As expected all backdoored models mainly focus on the triggers in backdoored inputs and the main objects in the clean ones.

## 4.5 STATE-OF-THE-ART DEFENSES

We now evaluate our attacks against the current state-of-the-art backdoor defenses. Backdoor defenses can be classified into two categories: *Data-based defenses* which detects if the input is backdoored, and *Model-based defenses* which detects if a model is backdoored.

**Model-based Defense:** We first evaluate all of our techniques in the multiple target label case against two of the current state-of-the-art model-based defenses, namely, Neural Cleanse (Wang et al., 2019) and ABS (Liu et al., 2019a). We start by evaluating the ABS defense. Here, we focus on the CIFAR-10 dataset since it is the only supported dataset by the published defense model. As expected, running the ABS model against our dynamic backdoored ones does not result in detecting any backdoor for all of our models. For Neural Cleanse, we use all three datasets to evaluate our techniques against it. Similar to ABS, all of our models are predicted to be clean models. Moreover, in multiple cases, our models had a lower anomaly index (the lower the better) than the clean model.

We believe that both defenses fail to detect our backdoors for two reasons. First, we break one of their main assumptions, i.e., that the triggers are static in terms of location and pattern. Second, we implement a backdoor for all possible labels, which makes the detection a more challenging task.

**Data-based Defense:** Next, we evaluate our three attacks against the state-of-the-art data-based defense, namely STRIP (Gao et al., 2019). We use all three datasets to evaluate the c-BaN models against this defense. First, we scale the trigger patterns by half while training the backdoored models, to make them more susceptible to changes. Second, for the MNIST dataset, we move the possible locations to the middle of the image to overlap with the image content, since the value of MNIST images at the corners are always 0. All trained scaled backdoored models achieve similar performance to the non-scaled backdoored models. Our backdoored

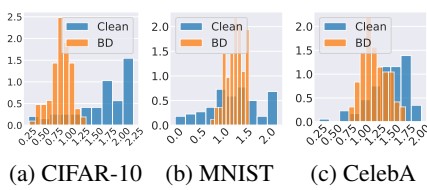

| (a) CIFAR-10 | (b) MNIST | (c) CelebA |

Figure 9: The histogram of the entropy of the backdoored vs clean input.

models successfully flatten the distribution of entropy for the backdoored data, for a subset of target labels. This subset of target labels makes picking a threshold to identify backdoored from clean data impossible without increasing the false positive rate, i.e., various clean images will be detected as backdoored ones. We visualize the entropy of our best performing labels against STRIP in Figure 9.

These results against both data and model based defenses show the effectiveness of our dynamic backdoor attacks, and opens the door for designing backdoor detection systems that work against both static and dynamic backdoors. We propose a data-based defense against the dynamic backdoor attack, however, due to space restrictions, we present it in Section A.3.

### 4.6 Evaluating Different Hyperparameters

We now evaluate the effect of different hyperparameters for our dynamic backdooring techniques. Due to space restrictions, we present here a summary of our findings and leave all the details in Section A.2. We start by evaluating the percentage of the backdoored data needed to implement a dynamic backdoor into the model. Our results show that using 30% is enough to get a perfectly working dynamic backdoor. Second, we evaluate the effect of increasing the size of the location set $\mathcal{K}$. Using more than four times the amount of possible locations, i.e., 16, still achieves the same performance. However, when we completely remove the location set $\mathcal{K}$ and consider all possible locations with a sliding window, the performance on both clean and backdoored datasets drops significantly. Finally, we evaluate the size of the trigger and the possibility of making it more transparent. For the trigger size, we find that a trigger with the size 5 is the smallest trigger than can achieve the best results. Similarly for the transparency, we find that setting the transparency scale to 0.5 or higher has no effect on the performance of the backdoor attack. However, when the scale is set below 0.5, the attack success rate starts degrading. To circumvent this limitation and make the triggers more transparent, we increase the trigger size. Our experiments show that when increasing the trigger size to the input images' size, our attacks can be performed with a scale of 0.1 with a negligible effect of the performance.

### 4.7 Relaxing the Threat Model (Transferability of the Triggers)

For our dynamic backdoor attacks, we assume the adversary to control the training of the target model. We now relax this assumption by only allowing her to poison the dataset.

First, it is important to mention that our Random Backdoor technique does not need to change the training of the target model, i.e., the adversary only needs to poison the training dataset with backdoored images and the corresponding target labels. Second, for both the BaN and c-BaN techniques, the adversary can rely on pre-trained BaN and c-BaN models instead of training them jointly with the target model. In detail, the adversary uses the pre-trained BaN and c-BaN model to generate multiple triggers and randomly place them to a set of – randomly picked – images. Then, she poisons the training set with this set of backdoored images and their corresponding target labels.

We use the MNIST dataset for evaluation and follow the same target models' structure as previously introduced in Section 4.3 and Section 4.4. However, to show the flexibility of our techniques, we use data from different distribution to pre-train the BaN and c-BaN models. We first use the CIFAR-10 dataset to train backdoored models with the BaN (Section 3.3) and c-BaN (Section 3.4) techniques. Next, we use the pre-trained BaN and c-BaN models to generate the backdoored dataset and poison the target dataset as previously mentioned. It is important to mention that the CIFAR-10 based BaN and c-BaN models generate 3-channel triggers, to use them to poison the MNIST dataset, we convert them to 1-channel triggers by taking the mean over the different channels. Finally, we use the poisoned dataset to train the target model.

As expected, the backdoored models achieve a perfect attack success rate (100%), while keeping the same utility as the backdoored models jointly trained with the BaN and c-BaN. This shows the flexibility of our attacks, i.e., the training procedure can be adapted by the adversary depending on her specific application. However, it is important to mention that jointly training the models has the advantage of giving the adversary more power, e.g., she can add a customized loss function to the target model while implanting the backdoor.

## 5 Conclusion

Current backdoor attacks only consider static triggers in terms of patterns and locations. In this work, we propose the first set of dynamic backdoor attack, where the trigger can have multiple patterns and locations. To this end, we propose three different techniques. Our first technique Random Backdoor samples triggers from a uniform distribution and places them at random locations of an input. For the second technique (BaN), we propose a novel generative network to construct triggers. Finally, we introduce c-BaN to generate label specific triggers. We evaluate our techniques using three benchmark datasets. The result shows that all our techniques can achieve almost a perfect backdoor success rate while preserving the model's utility. Moreover, we show that our techniques successfully bypass state-of-the-art defense mechanisms against backdoor attacks.

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

## A  APPENDIX

### A.1  MODELS ARCHITECTURE

For the target models' architecture, we use the VGG-19 (Simonyan & Zisserman, 2015) for the CIFAR-10 dataset, and build our own convolution neural networks (CNN) for the CelebA and MNIST datasets. More concretely, we use 3 convolution layers and 5 fully connected layers for the CelebA CNN. And 2 convolution layers and 2 fully connected layers for the MNIST CNN. Moreover, we use dropout for both the CelebA and MNIST models to avoid overfitting.

For BaN, we use the following architecture:

*Backdoor Generating Network (BaN)'s architecture:*

$$z \rightarrow \texttt{FullyConnected(64)}$$
$$\texttt{FullyConnected(128)}$$
$$\texttt{FullyConnected(128)}$$
$$\texttt{FullyConnected(|t|)}$$
$$\texttt{Sigmoid} \rightarrow t$$

Here, `FullyConnected(`$x$`)` denotes a fully connected layer with $x$ hidden units, $|t|$ denotes the size of the required trigger, and `Sigmoid` is the Sigmoid function. We adopt ReLU as the activation function for all layers, and apply dropout after all layers except the first and last ones.

For c-BaN, we use the following architecture:

*conditional Backdoor Generating Network (c-BaN)'s architecture:*

$$z, \ell \rightarrow 2 \times \texttt{FullyConnected(64)}$$
$$\texttt{FullyConnected(128)}$$
$$\texttt{FullyConnected(128)}$$
$$\texttt{FullyConnected(128)}$$
$$\texttt{FullyConnected(|t|)}$$
$$\texttt{Sigmoid} \rightarrow t$$

The first layer consists of two separate fully connected layers, where each one of them takes an independent input, i.e., the first takes the noise vector $z$ and the second takes the target label $\ell$. The outputs of these two layers are then concatenated and used as an input to the next layer (see Section 3.4). Similar to BaN, we adopt ReLU as the activation function for all layers and apply dropout after all layers except the first and last one.

### A.2  EVALUATING DIFFERENT HYPERPARAMETERS

Here we present the complete details on evaluating the different hyperparameters for our backdoor attacks.

**Proportion of the Backdoored Data:** We start by evaluating the percentage of backdoored data needed to implement a dynamic backdoor in the model. We use the MNIST dataset and the c-BaN technique to perform the evaluation. First, we construct different training datasets with different percentages of backdoored data. More concretely, we try all proportions from 10% to 50%, with a step of 10. In this settings, 10% means that 10% of the data is backdoored, and 90% is clean. Our results show that using 30% is already enough to get a perfectly working dynamic backdoor, i.e., the model has a similar performance like a clean model on the clean dataset (99% accuracy), and 100% backdoor success rate on the backdoored dataset. For any percentage below 30%, the accuracy of

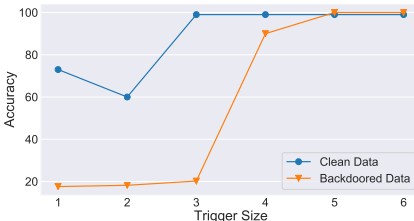

Figure 10: The result of trying different trigger sizes for the c-BaN technique on the MNIST dataset. The figure shows for each trigger size the accuracy on the clean and backdoored testing datasets.

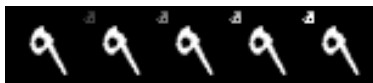

Figure 11: An illustration of the effect of using different transparency scales (from 0 to 1 with step of 0.25) when adding the trigger. Scale 0 (the most left image) shows the original input, and scale 1 (the most right image) the original backdoored input without any transparency.

the model on clean data is still the same, however, the performance on the backdoored dataset starts degrading. This demonstrates the ability of the adversary to implement a dynamic backdoor attack with 30% overhead for each target label, compared to training a clean model.

**Number of Locations:** Second, we explore the effect of increasing the size of the set of possible locations ($\mathcal{K}$) for the c-BaN technique. We use the CIFAR-10 dataset to train a backdoored model using the c-BaN technique, but with more than double the size of $\mathcal{K}$, i.e., 8 locations. The trained model achieves similar performance on the clean (92%) and backdoored (100%) datasets. We then doubled the size again to have 16 possible locations in $\mathcal{K}$, and the model again achieves the same results on both clean and backdoored datasets. We repeat the experiment with the CelebA datasets and achieve similar results, i.e., the performance of the model with a larger set of possible locations is similar to the previously reported one. However, when we try to completely remove the location set $\mathcal{K}$ and consider all possible locations with a sliding window, the performance on both clean and backdoored datasets drops significantly.

**Trigger Size:** Next, we evaluate the effect of the trigger size on our c-BaN technique using the MNIST dataset. We train different models with the c-BaN technique, while setting the trigger size from 1 to 6. We define the trigger size to be the width and height of the trigger. For instance, a trigger size of 3 means that the trigger is $3 \times 3$ pixels.

We calculate the accuracy on the clean and backdoored testing datasets for each trigger size, and show our results in Figure 10. Our results show that the smaller the trigger, the harder it is for the model to implement the backdoor behaviour. Moreover, small triggers confuse the model, which results in reducing the model's utility. As Figure 10 shows, a trigger with the size 5 achieves a perfect accuracy (100%) on the backdoored testing dataset, while preserving the accuracy on the clean testing dataset (99%).

**Transparency of the Triggers:** Finally, we evaluate the effect of making the trigger more transparent. More specifically, we change the backdoor adding function $\mathcal{A}$ to apply a weighted sum, instead of replacing the original input's values. Abstractly, we define the weighted sum of the trigger and the image as: $x_{bd} = s \cdot t + (1 - s) \cdot x$, where $s$ is the scale controlling the transparency rate, $x$ is the input and $t$ is the trigger. We implement this weighted sum only at the location of the trigger, while maintaining the remaining of the input unchanged.

We use the MNIST dataset and c-BaN technique to evaluate the scale from 0 to 1, with a step of 0.25. Figure 11 visualizes the effect of varying the scale when adding a trigger to an input.

Our results show that our technique can achieve the same performance on both the clean (99%) and backdoored (100%) testing datasets, when setting the scale to 0.5 or higher. However, when the scale is set below 0.5, the performance starts degrading on the backdoored dataset but stays the

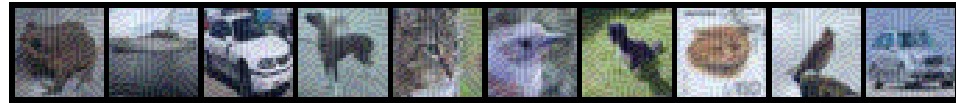

Figure 12: Visualization of the c-BaN backdoored images when setting the transparency scale to 0.1.

same on the clean dataset. We repeat the same experiments for the CelebA dataset and find similar results.

We believe that the transparency of our triggers can be further increased when using triggers with larger sizes. To this end, we use the CIFAR-10 dataset to repeat the experiments previously mentioned in this section. However, we set the trigger size to be the size of the image. Our experiments show that in this setting, our dynamic backdoor attack can still achieve a perfect attack success rate (100%) with a negligible drop in utility (0.3%). More concretely, the model's accuracy on clean data is 91.7% compared to the 92% accuracy of the backdoored model trained without any transparency. We visualize a set of randomly backdoored samples in Figure 12. As the figure shows, setting the scale to 0.1 makes the triggers hardly visible.

### A.3 POSSIBLE DEFENSES

Next, we propose a data-based defense against the dynamic backdoor attack. Intuitively, we use a denoising mechanism to filter triggers (as they can be considered as anomalies/distortions) out of the backdoored inputs. To this end, we use one of the most common denoising mechanisms, namely autoencoder. It works as follows: First, we train an autoencoder on clean data. Then, we use this autoencoder to reconstruct/denoise the inputs (by encoding then decoding them). The noise or triggers in our case are expected to be filtered out of the inputs due to two main reasons: The overfitting of the autoencoders to clean data and the lossy reconstruction process.

To implement our defense, we use the autoencoder to denoise all inputs before forwarding them to the target model. The autoencoder is expected to remove the trigger from backdoored data, while not significantly changing the clean ones.

To evaluate the efficacy of our proposed defense, we test it against the c-BaN technique, using both the MNIST and CIFAR-10 datasets. As expected, the backdoored images are not perfectly reconstructed by the autoencoder, i.e., the autoencoder does not fully reconstruct triggers. Our experiments show that in simple datasets like MNIST, our approach can successfully defend against the backdoor attack, with negligible utility loss (less than $1\%$). However, for more complicated datasets like CIFAR-10, the performance of our defense degrades. This is due to the high amount of details which hardens the reconstruction process of complex datasets (for both clean and backdoored inputs). For instance, the accuracy of the target model drops by $4.8\%$ and $25\%$ for the clean and backdoored dataset, respectively.

Another possible defense approach is first calculating the distance between the reconstructed input and the original input, then taking the decision to forward the input or not to the model, based on a predetermined threshold. We plan to explore this approach and other potential methods in future work.

