# OpenReview forum: "Dynamic Backdoor Attacks Against Deep Neural Networks"
_ICLR.cc/2021/Conference — Reject_

### Official Review · AnonReviewer1 · 2020-10-23
**Interesting idea, questionable threat model, poor literature review**

**Rating:** 5
**Confidence:** 4

**Review:**

Summary:
This paper proposed a class of methods for dynamic backdoor attack. The main idea is to generate different backdoor patterns and locations in backdoor attack. The threat model is the attacker has full access to the training data and the model training procedure. Both single-target and multi-target class-conditional triggers were explored. Experimental results on three datasets in the 100% poisoning setting verified the effectiveness of the proposed attack.

Strengths:
1. This paper investigated an important problem for secure deep learning.
2. The proposed methods were well motivated.
3. Empirical results indicate that dynamic backdoor triggers are possible.

Weaknesses:
1. Missing many existing works on backdoor attacks. The related work section only mentioned two backdoor attacks Gu et al., 2017 and Liu et al. 2019b. To my knowledge, there are many existing works in this field.
2. The discussion of the key properties for backdoor triggers are weak. For example, the injection rate, and the stealthiness of the trigger pattern. Existing works have proposed reflection triggers or even invisible triggers.  Compared to these triggers, the proposed triggers are not stealthy at all. Appearing in different places of the input image even makes the trigger more easily detectable (at least to human checkers).
3. The proposed methods need to poison at least 30% of the training data, however, many existing methods only poison 10%-20% of the training data.
4. The 6 training steps described on Page 6 indicate that the proposed methods need to modify the model training process. This is different from a typical backdoor setting where the attacker cannot control the training procedure, which allows the attacker to invade commercial models by only poisoning the data. From this perspective, the setting adopted in this paper is questionable.
5. The 30% poisoning setting in Section 4.6 is not clear. Was the 30% only used for training the BaN then poisoning 30% data using the BaN model or does it still need the entire training data (but only 30% contain the trigger)? The training process on Page 6 says the BaN and the backdoored model need to be trained simultaneously (step 6).
6. One more concern, the training of BaN nets needs a certain proportion of the training data. Even 30% is still a lot for large-scale datasets like ImageNet. However, many existing attacks do not need a generator, which makes the attack more applicable and convenient. Can the generators trained on one dataset transferable to other datasets?

---
Comments after the rebuttal:

Overall, the attack settings are still questionable: low stealthiness, high poisoning rate, and intervening the training process.

- The authors clarified that "Neither our random backdoor technique nor the transferred BaN and c-BaN requires that", however, in 3.1 Threat Model, it is clearly stated "The dynamic backdoor attack is a training time attack, i.e., the adversary is the one who trains the backdoored DNN model. To implement our attack, we assume the adversary controls the training of the target model and has access to the training data following previous works (this is actually not true)", the authors seem misunderstood the different between "controlling the training of the model" to data poisoning.

In terms of the difference to static trigger,

- The authors argued that they are different to static trigger and "a static trigger cannot be used as a dynamic trigger", which I don't find it is verified somewhere. Actually the random backdoor proposed in this paper proves that a static trigger when applied to a random location still works very well, even as good as BaN and c-BaN. Whether existing triggers can already achieve a dynamic effect or not is unclear, or verified.

Without knowing how hard it is to make a static trigger works in a dynamic manner, or where it will completely fail, it is hard to evaluate the contribution and novelty of the proposed attacks. In other words, it seems that a static trigger attached to random locations is good enough to achieve the dynamic purpose, which I don't think is contributive enough.

Stealthiness and poisoning rate is another concern, say when you apply the BadNets to random locations, how high the poisoning rate will be to achieve 100% ASR? Higher than 20% or need 100%?

Overall, the lack of thorough comparisons of the trigger properties to existing attacks is the major weekness.

So my rating will stay the same.

---

> ### Author Response · Authors · 2020-11-20
> **Response to Reviewer 1**
>
> 1- We updated the related work section to cover more of the current state-of-the-art backdoor attack works.
>
> 2- We agree with this point. We added to our paper an experiment to make the triggers more stealthy in the appendix A.2. However, we would like to mention that the main purpose of our work is not to make the triggers more stealthy, but to have a dynamic behavior.  We choose to have visible patterns as the triggers to make the attack possible in the physical world, i.e., sticking a pattern on the target object to misclassify it.
>
> 3- We also agree with this point. We believe our attacks need a higher percentage as the adversary needs to train the c-BaN or BaN models. However, when using a pre-trained BaN or c-BaN model, the percentage can be dropped to below 20%.
>
> 4- We added a new section (Section 4.7) to show how to relax this assumption and implement the dynamic backdoor attack with only poisoning the training data (intuitively, we use pre-trained BaN and c-BaN models).
>
> 5- The 30% means that only 30% of the data is considered when training the BaN. Or in other words, the BaN model is only updated by 30% of the date when training the target model; the rest of the data is used to update the target model.
>
> 6- Yes, as we show in the newly added section (Section 4.7), the trigger generators can transfer to different datasets with different data distributions.

---

> > ### Comment · AnonReviewer1 · 2020-11-25
> > **Thanks for the clarification**
> >
> > Thanks for the response. The transfer study is helpful, however, it should be supported by more detailed results rather than just saying it is 100% transferable. Overall, the attack settings are still questionable: low stealthiness, high poisoning rate, and intervening the training process. For the stealthiness, the authors argued that visible  patterns are important for physical world attack. I agree with this point. However, isn't that static triggers become dynamic itself in the physical world scenario? For example, when a sticker is attached to an object, photos taken from various angles will end up with the trigger pattern appears at different places of the image. In this case, why the proposed dynamic trigger is superior to static patterns? And also, the claim that visible triggers are more effective is not evidenced, has this been analysed in any existing works?
> >
> > ---
> > I will keep my rating as it is.

---

> > > ### Author Response · Authors · 2020-11-25
> > > **Response to Reviewer 1**
> > >
> > > We thank the reviewer for the additional feedback.
> > >
> > > --  More experiments on transferring study: Due to the time limit, we only show the transferring attack from CIFAR-10 to MNIST. However, we believe other setups will have similar results and plan to evaluate and include them in the next version of the paper.
> > >
> > > -- Intervening the training process: Neither our random backdoor technique nor the transferred BaN and c-BaN requires that. All the adversary needs to do is poisoning the data.
> > >
> > > --  High poisoning rate: We empirically found that the transferred attacks and random backdoor only need to poison less than 20% of data.
> > >
> > > --  Stealthiness and comparison to static backdoor: To launch our attack, the procedure is similar to the static backdoor, that is adding a trigger to one part of the image. However, the current static backdoor only allows the attacker to add a fixed trigger on a fixed location of the image.  For instance, in the given example, for the static backdoor to work, the angle should be fixed so that the trigger is in the correct position. If the angle changes and the trigger appears in a different position, the backdoor will not be activated. Hence, a static trigger cannot be used as a dynamic trigger. More importantly, static backdoor attacks can be effectively detected by the current defense mechanisms. On the other hand, our attack allows the attacker to find a dynamic pattern trigger and add it to a dynamic position on the image. This makes our dynamic backdoor different from the static backdoor.
> > >
> > > -- Transparency: To our knowledge, current backdoor attacks do not use transparent triggers when implementing the attack in the physical world. This is indeed an interesting direction, and we plan to explore it in the future with lab study.

---

### Official Review · AnonReviewer2 · 2020-10-28
**Is it possible to improve backdoor attacks to human perception (say via steganography?)**

**Rating:** 6
**Confidence:** 3

**Review:**

Summary:

This paper outlines improved backdoor attacks for deep neural networks. Backdoor attacks are training-time attacks whereby an adveresary trains a network in such a way that it functions as a classifier on honestly generated images, but it has the added caveat of also being able to misclassify on images corrupted via a "trigger" set by an adversary. This is a grave security concern, and as such there is substantial literature that relates to both systematically training backdoored DNNs, and also detecting when a trained DNN has a backdoor.

As the authors mention, the existing literature on backdoors typically has a static backdoor where a single portion of an image is altered to create a trigger in a deterministic fashion / location. The authors improve the functinoality of backdooring a DNN by creating a training process where triggers can now appear at random locations in the image and can have random contents. The way they do so is by judiciously and simultaneously training a GAN in conjunction to the DNN that will be corrupted. The GAN uses the DNN under training as a discriminator and ultimately produces corrupted images with triggers that are meant to push the DNN towards having a backdoor through typical training procedures. These GAN-based techniques are key to both algorithms the authors present: BaN (random location triggers) and c-BaN (random location and contents to trigger).

Finally, the authors validate how effective their backdoor techniques are by training on MNIST, CelebA and CIFAR-10 datasets. Their backdoored DNN have their performance measured by how well they misclassify on triggered images and on how little they compormise on state of the art accuracy in non-backdoored models. The authors show that their models have a small sacrifice in terms of accuracy (although their model interestingly enough classifies better on CelebA), and furthermore, their backdoors are robust to existing methods used to catch backdoored DNNs


Pros:
I was not as familiar with existing literature on backdoored DNNs until reading this paper. I see how this is a valid concern in the community, and I believe that the authors have provided a better attack with respect to the state of the art. Furthermore, their simulations do a good job of providing evidence of this improvement (in terms of both the visual output of the triggers, and also the performance against software used to find backdoored DNNs)

Questions / Points to address:
-As someone who is not as conversant with relevant literature on backdoored DNNs, I wonder how much work has been done on the fact that the triggers used are easily identifiable by a human looking at the picture (in termss of a blatant square in a given location). Though I am also no expert, there is a mature field of Steganography that focuses on mathematical methods of concealing information in images (and other media files). Is there significant overlap in what has been studied with this? Can the DNN architecture used be informed by existing theory in this field?
-I am also curious about the fact that the backdoored classifierse performed better on CelebA vs. other data sets. I see the authors' point about this possibly being a form of regularisation, but it would be interesting to see the performance of this methodology on more complex datasets that may also exhibit this behaviour.

---

> ### Author Response · Authors · 2020-11-20
> **Response to Reviewer 2**
>
> Thank you for your insightful comments and feedback; we address each of the points raised independently below:
>
> 1- This is a very interesting insight. To the best of our knowledge, there are currently no works that use steganography to implant backdoors. Thank you for this idea, we plan to study it in future work.
>
> 2-  We will further inspect this observation. According to our current experiments, the more generalized the target model, the less the chance of this to happen, i.e., the backdoored models will have either equal or worse performance.

---

### Official Review · AnonReviewer5 · 2020-11-05
**review of dynamic backdoor**

**Rating:** 5
**Confidence:** 4

**Review:**

Summary and contributions:
In this work, the authors propose the first set of dynamic backdoor attacks, in terms of trigger pattern and location. Three techniques which areRandom Backdoor, Backdoor Generating Network (BaN) and conditional BackdoorGenerating Network (c-BaN) are used to implement dynamic backdoor attacks. The results of empirical evaluation show that the proposed attacks are effective as well as difficult to defend by several defense mechanisms.

Strength (significance and novelty):
1. They propose the first class of dynamic backdoor attacks. And this work is well motivated. Previous backdoor attacks only use a fixed and static backdoor trigger, which is set to the default assumption of several defense mechanisms. Dynamic backdoors chosen from a set of predefined triggers and locations or generated byGAN, can bypass these defense mechanisms.
2.  In the empirical evaluation, the proposed attack algorithm is effective varying different datasets and is robust to several defense mechanisms.
3. In addition to single target label and multiple target label attacks, they also propose an auto-encoder based defense against dynamic backdoor attacks.
4. The techniques in dynamic backdoor attacks are simple yet effective. And the paper is well-written and easy to follow.

Weakness:
1. This work only considers white-box backdoor attacks, i.e., the attacker controls the target model and has access to the training data. I suggest the authors should refer Refool [1] and study the transferability of BaN and c-BaN to different data distributions and model architectures.
2. This work aims to improve backdoor robustness to defense mechanisms. However, theSec 4.5 state-of-the-art defenses is too brief. More experiments and illustrations should be added to this part. Here are some suggestions:

3. Since the motivation of this work is to “break” the assumption in some model-based defenses, it would be interesting to compare the reversed triggers and the original triggers (sample from the dynamic trigger distribution). In addition to Neural Cleanse and ABS, MESA [2] can also be considered for its superior to Neural Cleanse.

4. Add some visualizations to understand dynamic backdoor attacks, e.g. Grad-CAM,  which is widely adopted in backdoor attacks [1, 3].

5. Add a comparison with traditional static backdoor attacks against different defense mechanisms, which would be more convincing.

Reference
1. Liu, et al. Reflection Backdoor: A Natural Backdoor Attack on Deep Neural Networks.ECCV 2020.
2. Qiao, et al. Defending Neural Backdoors via Generative Distribution Modeling. NeurIPS2019.
3.  Xie, et al. DBA: Distributed Backdoor Attacks against Federated Learning. ICLR 2020.

---

> ### Author Response · Authors · 2020-11-20
> **Response to Reviewer 5**
>
> Thank you for your insightful comments and feedback; we address each of the points raised independently below:
>
> 1- We updated the related works section, to better reflect the cover more of the current state-of-the-art backdoor attack works, including Refool. We also added a new section (Section 4.7) to show the transferability of our attacks. Using this transferability property, the adversary can implement the dynamic backdoor attack with only poisoning the training data.
>
> 2- MESA is a very interesting approach, however, they assume the knowledge of the trigger size and only publish the code for fixed trigger sizes.  Due to time constraints, we cannot reimplement their code, however, when running the code against our models. It fails to detect any backdoors. But it has to be mentioned that our triggers have the size of 5x5, and their code assumes 3x3.
>
> 3- We added visualization using Grad-CAM in Section 4.4.
>
> 4- We agree to this point. However, we refer - due to space restrictions - to the results in the defense papers as they evaluate their works against the static backdoor attacks.

---

### Decision · Program_Chairs · 2021-01-07
**Final Decision**

**Decision:**

Reject

**Comment:**

The authors present a new set of trigger based backdoor attacks that use dynamic patterns that make detection harder. These attacks seem to be stronger with regards to state of the art attacks.

Some weaknesses is the need for full whitebox access of the model. Several key references are missing, and the comparison with other backdoor attacks in unclear.

Moreover, although the authors compare with trigger based backdoors, there are plenty of triggerless backdoors that can be viewed as dynamic, eg the attacks by
Bagdasaryan et al. http://proceedings.mlr.press/v108/bagdasaryan20a.html and works referencing this paper.

The paper indeed provides an interesting path towards dynamic attacks, but the lack of comparisons with state of the art literature, and also the need for whitebox access/high poisoning rate significantly limit the novelty of this work.